# A Prospective Study of Longitudinal Risks of Cognitive Deficit for People Undergoing Glioblastoma Surgery Using a Tablet Computer Cognition Testing Battery: Towards Personalized Understanding of Risks to Cognitive Function

**DOI:** 10.3390/jpm13020278

**Published:** 2023-01-31

**Authors:** Rohitashwa Sinha, Riccardo Masina, Cristina Morales, Katherine Burton, Yizhou Wan, Alexis Joannides, Richard J. Mair, Robert C. Morris, Thomas Santarius, Tom Manly, Stephen J. Price

**Affiliations:** 1Department of Neurosurgery, Addenbrooke’s Hospital, Cambridge CB2 0QQ, UK; 2Department of Clinical Neurosciences, University of Cambridge, Cambridge CB2 0QQ, UK; 3Leeds Institute of Medical Research, University of Leeds, Leeds LS9 7TF, UK; 4Department of Oncology, Addenbrooke’s Hospital, Cambridge CB2 0QQ, UK; 5MRC Cognition and Brain Sciences Unit, Cambridge CB2 7EF, UK

**Keywords:** glioblastoma, cognition, surgery, deficits, risks, quality of life, survivorship, neuro-oncology

## Abstract

Glioblastoma and the surgery to remove it pose high risks to the cognitive function of patients. Little reliable data exist about these risks, especially postoperatively before radiotherapy. We hypothesized that cognitive deficit risks detected before surgery will be exacerbated by surgery in patients with glioblastoma undergoing maximal treatment regimens. We used longitudinal electronic cognitive testing perioperatively to perform a prospective, longitudinal, observational study of 49 participants with glioblastoma undergoing surgery. Before surgery (A1), the participant risk of deficit in 5/6 cognitive domains was increased compared to normative data. Of these, the risks to Attention (OR = 31.19), Memory (OR = 97.38), and Perception (OR = 213.75) were markedly increased. These risks significantly increased in the early period after surgery (A2) when patients were discharged home or seen in the clinic to discuss histology results. For participants tested at 4–6 weeks after surgery (A3) before starting radiotherapy, there was evidence of risk reduction towards A1. The observed risks of cognitive deficit were independent of patient-specific, tumour-specific, and surgery-specific co-variates. These results reveal a timeframe of natural recovery in the first 4–6 weeks after surgery based on personalized deficit profiles for each participant. Future research in this period could investigate personalized rehabilitation tools to aid the recovery process found.

## 1. Introduction

Glioblastoma, the most aggressive primary brain cancer, has a median overall survival of less than two years [1,2] and is associated with a considerable burden of cognitive deficits [3,4,5,6,7,8] due to both the diffuse nature of the disease [9] and the aggressive nature of treatment with maximal surgical resection [10] and radical radiotherapy regimens [11]. Cognitive function has been associated with poor health-related quality of life and the subject of patient-reported outcome literature, in particular a comprehensive survey of 1004 patients by the Brain Tumour Charity that described what life is really like for adults living with a brain tumour [12,13,14,15,16].

Much of the literature regarding cognitive deficits in the perioperative period is of limited generalizability to patients with glioblastoma due to heterogeneous sampling of patients with glioblastoma alongside patients with less aggressive brain tumours and considerable data loss from longitudinal participant drop-out [17]. Some studies that have focused on participants with glioblastoma have assessed cognition at 3 months after surgery [8,18,19]. This is usually when participants have already started adjuvant treatment with chemotherapy and radiotherapy, making it more difficult to understand the associations of surgical resection for glioblastoma on participants’ cognitive function.

The aim of this current study is to address this knowledge gap by providing longitudinal data about risks to cognitive function in participants undergoing surgery for glioblastoma, from before surgery to after surgery but before any adjuvant treatment. Such data may help clinicians and patients in making more informed and personalized choices about the surgical resection treatment offered when first diagnosed with a glioblastoma. 

Our primary study objective is to use an electronic cognitive test battery, ‘OCS-Bridge’ (https://ocs-bridge.com/ (accessed on 26 January 2023)), to assess patients with glioblastoma before and after surgery for cognitive deficits and analyse the odds risks ratios for specific cognitive functions tested by the battery. We hypothesize that surgical resection will exacerbate cognitive deficits detected before the surgery in these patients diagnosed with glioblastoma. A secondary data sharing objective is to contribute our dataset with detailed annotation, including cognitive data, clinical metadata, and demographic information for future synthesis in meta-analysis research.

## 2. Materials and Methods

### 2.1. Study Design

We performed a prospective, longitudinal, observational cohort study. To detect longitudinal cognitive deficits in glioblastoma patients from before to after surgery, at 80% power and 5% significance, we used heathy control means derived from a normative population of 300 volunteers who had previously taken the tests in the OCS-Bridge battery, using parallel sessions of the tests that were validated in previous studies [20,21,22,23]. This normative healthy control data provided the between-participant and test–retest reliability for automated comparison of participant performance (https://ocs-bridge.com/ (accessed on 26 January 2023)). We used the OCS-Bridge cognitive testing battery in these studies, as it shares the advantages of other computerized cognitive testing batteries of using electronic versions of well validated neuropsychological tests, broadening access by removing the need for specialist neuropsychology trained staff to deliver the tests and providing additional test metrics, such as response times in milliseconds. Test results were automatically calculated into categories based on these normative performances, indicating normal performance or impaired performance (at 2 SD and having 0.05 probability). Effect sizes (Cohen’s D) ranged between 0.25–6.27, median 1.3, and IQR 0.6–3.3, which are comparable to other clinical-versus-healthy population cognitive screening instruments and are consistent with our previous research using these methods [21,24]. The sample size calculation for this within-subjects analysis stipulated 40 patients, where the probability of new deficit is at least 30% and the probability of recovery from deficit is greater than 5%. All longitudinal data were analysed using paired sample tests to avoid pseudoreplication. 

### 2.2. Participants

We recruited adult patients admitted to Addenbrooke’s hospital between May 2017 and January 2020 in an ethically approved study investigating longitudinal cognitive function in patients with glioblastoma (Harrow Research Ethics Committee: 18/LO/0491). Written informed consent was obtained from all patients. All procedures were in accordance with the Declaration of Helsinki. Inclusion criteria were (i) diagnosis of glioblastoma, (ii) intended surgical removal of at least 90% of the enhancing tumour, (iii) suitable for subsequent radiotherapy (60 Gray) with concomitant Temozolomide, (iv) World Health Organization performance status of zero or one, and (v) intact capacity for longitudinal cognitive testing before and after surgery. A sampling strategy of recruiting all consecutive potential participants that met the inclusion criteria was employed. 

### 2.3. Neuropsychological Assessment

These were performed using tests in the tablet-based screening battery OCS-Bridge (https://ocs-bridge.com/ (accessed on 26 January 2023)) during ‘presurgical-assessment’ visits (A1), postoperatively on the day of hospital discharge or before the first clinic review thereafter (A2), and for patients returning to Addenbrooke’s for adjuvant treatment with radiotherapy at 4–6 weeks after surgery before the start of adjuvant treatment (A3). Of note, the A3 tests were not undertaken for all recruited participants that had A1 and A2 assessments. Where all participants had their tumour debulking surgery at Addenbrooke’s hospital and hence were available for the A1 and A2 assessments, some of these returned to other hospitals closer to their homes for adjuvant treatment and hence missed the A3 assessments.

All tests within the OCS-Bridge battery were administered to all patients (Appendix A). Anxiety and mood were assessed with Generalised Anxiety Disorder-7 and Patient Health Questionnaire-9 [25,26]. Postoperative testing used parallel tests in OCS-Bridge and scores were corrected against normative control data for reliable change [27]. 

### 2.4. Variables

The individual cognitive tests classified into cognition domains had varying distributions (severely skewed and normally distributed), multi-collinearity, and varying data granularity. In order to homogenise the analysis across these varying tests within the cognitive domains, the main analyses were based on binary categorical data outcomes of intact cognition or performance consistent with deficit when compared to the healthy normative scores.

The gross ‘location’ of the contrast enhancing bulk of the tumour was determined by hemisphere and lobe. The lobar locations were classified into frontal, temporal, and parietal-occipital lobes, where parietal and occipital were grouped together due to their small cortical volumes by comparison to frontal and temporal and the common finding of tumours in this region straddling both lobes rather than being ‘primarily located’ in either. These location details were taken from the clinical records of reports by Consultant Neuroradiologists. Where reports stated tumour bulk was in two lobes, the lobe with the majority of the tumour was recorded as its ‘location’.

Clinical co-variates of potential confounding effect were included in the analysis:Demographic data: age [28] and sex, years of education [29];Anxiety [30] and depression [31];Medications: dexamethasone steroid [32] and anti-epileptic medications [33] given peri-operatively as per electronic record charts. The highest recorded dose of each was used as a surrogate measure for each to account of the effect size of this variable;Molecular marker information: IDH and MGMT status [34,35];Surgical adjuncts: the use of 5-Aminolevulinic acid [10], intraoperative neurophysiology or awake mapping [36];Surgical outcome: complete resection of enhancing tumour or subtotal with residual [37];Timing of assessment: all dates of surgery and subsequent assessments were recorded in number of days from the first assessment to determine whether there was a recovery time effect [38];Tumour Volume: calculated using the measurements from the reports by Consultant Neuroradiologists for consistency (perpendicular dimensions in the maximal axial plane slice of MRI) [35,39].

### 2.5. Statistical Analysis

Cohort characteristics have been described using means and interquartile range (IQR), or counts and frequencies, as appropriate. All *p*-values are reported as unadjusted 2-sided *p*-values unless stated otherwise. The commonly accepted threshold of <0.05 was adopted for statistical significance. Patients with missing data were excluded from matched A1–A2 and A2–A3 analysis. All statistical analyses were performed in R using the epitools and exact 2 *×* 2 software packages. The authors opted to deal with the issue of multiple comparison by transparently reporting all results regardless of their statistical significance, as recommended by Perneger [40]. Since it is not possible to determine the true number of comparisons performed in this analysis and the variables exhibit multicollinearity, it would not be appropriate to arbitrarily adjust for multiple comparison. Therefore, all *p*-values reported in this study are unadjusted. For data which include moderate and conservative multiple comparisons corrections applied, please refer to the Appendix A.

Potential associations between covariates and cognitive deficit were explored by logistic regression in both univariate and multivariate models with bidirectional stepwise evaluation of variables of interest.

#### 2.5.1. Assessment by Cognitive Test: Tumour vs. Normative Assessment

The rate of cognitive deficits between our cohort (A1, A2, and A3) and the normal population (N) were compared by unconditional maximum-likelihood estimation of odds-ratio (ad/bc) with Wald 95% confidence intervals. By N, we refer to a sample of 300 healthy individuals providing a normative sample with prevalence of cognitive deficit of the lowest 5–10% of scores (2 SD). Unless otherwise stated, all comparisons refer to the entire glioblastoma cohort at the specified timepoint. For subgroup analyses by hemisphere, lobe, and hemisphere–lobe, the OR of cognitive deficit was estimated both relatively to the normal population (e.g., temporal lobe vs. normal) and relatively to the other glioblastoma cases within the same timepoint (e.g., temporal lobe vs. other lobes at A1).

#### 2.5.2. Assessment by Cognitive Test: Tumour vs. Tumour Assessment

To assess the differences between timepoints A1–A2 and A2–A3, the observed frequencies of cognitive deficit were compared by McNemar’s test to account for matching. For these comparisons, 95% confidence intervals and *p*-values were estimated by exact statistics based on the binomial distribution on one of the off-diagonal values conditioned on the total of both off-diagonal values. 

#### 2.5.3. Assessment by Cognitive Domain

Analysis by cognitive domain was performed analogously to the analysis by a cognitive test. A cognitive domain was considered affected by deficit if at least one cognitive test within the domain was positive within the domain.

### 2.6. Data Availability

For data sharing and prevention of research waste, the raw data used to determine the odds risks presented are included as Appendix A.

## 3. Results

### 3.1. Study Participants and Data Collection

Overall, 49 patients were prospectively included at baseline pre-surgical assessment (A1). Due to four patients suffering complications such as stroke and seizures, the number of patients eligible to continue with testing was 45 at A2 in the early phase after surgery. Of these, 24 patients stayed to have adjuvant treatment in our unit and were tested before adjuvant treatment began at A3, 4–6 weeks after surgery.

The median number of days from A1 presurgical assessment to surgery was 1 (interquartile range: 1–5 days). The median number of days from surgery to A2 assessment was 3 days (interquartile range 2–8 days). The lowest of this A2 postoperative range was 2 days, as this was the minimal time when participants had recovered sufficiently from their general anaesthetic and were safely mobile enough to consider being discharged home.

Cohort characteristics are summarised in Figure 1, and provide a visual representation including demographics, cognitive performance, and patient-specific characteristics. The variance in cognitive deficit profiles per participant reflect the personalized impact of these losses of function.

### 3.2. Patients with Glioblastoma Have Impaired Cognitive Function in Several Domains before Surgery

A summary of the risk of deficit for each of the six cognitive domains assessed in our cohort is provided in Figure 2A. Before surgery (A1), patients with glioblastoma were characterized by an increased risk of cognitive deficit in 5 out of 6 cognitive domains compared to the general population. Of these, the risks of Attention (OR = 31.19, 95% CI = 17.76–86.45), Memory (OR = 97.38, 95% CI = 38.87–243.95), and Perception (OR = 213.75, 95% CI = 67.89–672.95) deficits were markedly increased. Notably, the risk of Praxis deficit at A1 was not significantly different to that of the general population (OR = 2.65, 95% CI = 0.98–7.2). Findings by individual cognitive test rather than by cognitive domain are summarised in Figure 2B.

Data are shown as OR (diamonds) ± 95% CI. OR > 1 indicates an association with increased risk of cognitive deficit relative to baseline, while OR < 1 indicates an association with reduced risk of cognitive deficit relative to baseline. Statistically significant differences between the risk of cognitive deficit at A1 and at A2 are marked by a ‘*’. Raw data are available as scatterplots in Appendix A.

We studied hemispheric differences by comparing the observed risks of cognitive deficits between participants who had contrast enhancing glioblastoma on MRI in the left and right cerebral hemispheres (Figure 2C). Interestingly, the risk of Praxis deficit was significantly greater in participants, with glioblastoma affecting the Left Hemisphere (OR = 5.7, 95% CI = 1.05–30.87) more than the Right Hemisphere (OR = 0.33, 95% CI = 0.02–7.13). The association between lobar location and cognitive deficit, as well as the combined effect hemisphere–lobe, is summarised in Figure 2D,E.

### 3.3. Cognitive Function Worsens in the Early Postoperative Period

While the risk of cognitive deficit was increased in 5/6 domains pre-operatively (A1), in the early postoperative period we observed an increased risk of deficit in 6/6 domains (A2, N = 45) (Figure 2A). Overall, the largest change from A1 affected Attention (A2: OR = 1606.16, 95% CI = 98.57–28502.48), followed by Praxis (A2: OR = 11.54, 95% CI = 5.21–25.56) and Language (A2: OR = 15.83, 95% CI = 7.20–34.83). Data relative to each individual test are available in Figure 2B.

When stratifying by hemisphere, we can see that Memory, Number, and Perception domains have similar risks of cognitive deficits at A1 and A2, irrespective of hemisphere. Hemispheric differences are seen in Praxis, for which cognitive function worsens significantly more for right-sided tumours (OR = 15.83, 95% CI = 5.9–42.48) than for left-sided tumours (OR = 8.31, 95% CI = 2.97–23.26). Attention worsens at A2 for participants with either left (OR = 865.71, 95% CI = 50.21–14,927.31) and right (OR = 828.87, 95% CI = 48.01–14,310.34) hemispheric lesions. However, for the group with lesions in the left hemisphere, this change is statistically significant using McNemar’s Exact test (Left *p*.value = 0.007, Right *p*.value = 0.06).

The effects of lobar location on cognitive deficit at A2 are depicted in Figure 2D. Of note, the risk of Attention deficit increases postoperatively in all lobar subgroups, but most markedly in the Temporal lobe (OR = 755.19, 95% CI = 43.61–13-76.47). Figure 2E summarises the combined effect of hemisphere–lobe on the risk of cognitive deficit.

### 3.4. Some Cognitive Function Recovers over Time in the Late Postoperative Period

For N = 24 patients, we were able to assess cognitive function in the late postoperative period of 4–6 weeks just before starting radiotherapy (A3). Considering the limited size of the cohort at A3, the general trend observed across all domains was an improvement towards preoperative risks of cognitive deficit in some domains (Figure 3). In the case of Praxis, there is weak evidence that the risk of deficit at T3 may even be lower than at A1 (A1: OR = 2.65, 95% CI = 0.50–13.93; A3: OR = 0.31, 95% CI = 0.01–6.83), though this was hindered by the limited size of the A3 cohort. When comparing the risk of cognitive deficit by tumour location by either hemisphere or lobe, no differences were observed between groups.

Data shown as OR (diamonds) ± 95% CI. OR > 1 indicate an association with an increased risk of cognitive deficit relative to healthy control baseline, while OR < 1 indicates an association with reduced risk of cognitive deficit relative to healthy control baseline. Raw data are available as scatterplots in Appendix A.

### 3.5. The Observed Risks of Cognitive Deficit Are Independent of Patient-Specific, Tumour-Specific, and Surgery-Specific Characteristics

We sought to assess whether any patient-specific, tumour-specific, and surgery-specific characteristics were linked with the observed deficits in cognitive function at A1, A2, and A3, thus confounding the associations described above. We tested the association of all covariates listed in Figure 1, both by univariate and multivariate analyses (Table 1 for A1 analysis and Table 2 for A1 to A2 analysis, Appendix A). Appendix A contain visual representations of the association between each covariate and cognitive deficit in each domain.

None of the covariates analysed were strongly associated with cognitive deficit at any timepoint within our cohort. At A1, none of the covariates were associated with risk of cognitive deficit. At A2, Age was weakly associated with perception and praxis deficit (Perception: OR = 1.15, 95% CI = 1.03–1.29; Praxis: OR = 1.15, 95% CI = 1.04–1.27), and IDH-positive status was associated with perception deficit (OR = 0.01, 95% CI = 0.00–0.28) (Appendix A). Both effects were lost in multivariate analysis (Table 2, and Appendix A).

Lastly, we tested for potential effects of covariates on the risk of developing new deficits between A1 and A2 within our cohort. Weak associations were found between Age and Praxis (OR = 1.09, 95% CI = 1.00–1.19), and years of education and Memory (OR = 1.45, 95% CI = 1.03–2.04) (Appendix A). Both associations were lost in multivariate analysis (Table 2, and Appendix A).

### 3.6. Risk Communication via Visual Representation

To facilitate the task of risk communication with patients in the clinical setting, we have provided the reader with two visual tools in line with risk communication recommendations [41] and the guidelines set by the National Institute for Clinical Excellence [42]. Appendix A consists of the same data as Figure 2 but converted into natural frequencies. Appendix A provides the reader with an opportunity to frame the risks that are being communicated within a broader spectrum of more common, intuitively understood risks.

## 4. Discussion

This is the first study to prospectively use electronic tablet computer cognitive testing in the early perioperative phase to demonstrate the high risks of accruing additional cognitive deficits in a cohort of patients exclusively with glioblastoma undergoing resection surgery. Compared with the performance of healthy controls on the same battery of tests, patients with glioblastoma already have very high risks of multiple domain cognitive deficits before surgery, and we have found these risks to increase even further in the early postoperative period. This is the period when patients are being discharged home from the admission to hospital for surgery and when they are receiving the histological diagnosis of glioblastoma for the first time.

We have also shown, in a subset of these patients who avoided complications and were eligible for maximal adjuvant treatment with chemotherapy and radiotherapy, that these risks of deficits can improve from the early post-surgical assessment period towards the risk level before surgery by the time that these patients are due to start adjuvant therapies 4–6 weeks after surgery. Furthermore, we have presented our data in multiple diagrammatic ways to aid pre-surgical counselling so that patients can, in the future, better understand the risks of undergoing resection surgery on their cognition. Finally, we have not found various clinical, radiological, histopathological, and treatment variables to have any association with the changes in risks to cognitive function shown.

This study has benefitted from the use of computerised cognitive testing to improve the compliance of study participants over traditional ‘pen-and-paper’ testing. Other advantages are the automated analysis of results instant comparison to normative healthy control performances on the same tests and broadening of access for participants with glioblastoma from not requiring specialist neuropsychology expertise to administer these tests. It is the increased flexibility from this that has led to these tests fitting into the early postoperative period between 3–10 days after surgery to demonstrate the early postoperative risks to cognitive function that have not been shown in other studies. In addition, by having two assessments in a subset of these patients before they start chemotherapy and radiotherapy, we can show a trend suggestive of the natural history of cognitive recovery after surgery, namely that the initial deterioration seen within the first 2 weeks can improve towards the baseline before surgery over 4–6 weeks from surgery. The high retention and completion of follow up assessment for those eligible compared with other studies is likely in part due to our participants preferring computerised format testing to traditional ‘pen-and-paper’ alternatives [43].

A final strength of the current study is the level of annotative detail included in the co-variate modelling, spanning from steroid and antiepileptic medication usage to radiological tumour volume and molecular diagnostic sub-typing based on IDH and MGMT promoter status. Whilst such confounding variables as years of education and usage of steroid and anti-epileptic medications have been associated with cognitive deficits and overall outcomes [32,33], in the current study the effect size of such interactions is much smaller than the time in relation to surgery and hence is insufficiently powered to detect them. Towards the end of the study, patients were still undergoing routine treatment for glioblastoma during the start of the COVID-19 pandemic. Whilst none of the patients in the study had a COVID-19 infection, we have included this important potential co-variate for those participants overlapping with the beginning of the pandemic and found no relationship on the risk of cognitive deficits.

We have presented our data in traditional formats for medical data dissemination such as forest plots, but also in more digestible formats commonly used to explain risks directly to patients and carers such as scaled pictographs and risk spectra as the general public has become more accustomed to considering risk information dissemination during the COVID-19 pandemic [44]. By presenting the risk data using visual aids and natural frequencies rather than simply numbers and percentages, these can be used as tools to aid personalised decision-making processes between patients and clinicians as recommended by the National Institute for Clinical Excellence [42]. Given that the cognitive deficit risks are already high before surgery when surgery is discussed, these tools may be particularly important to facilitate shared decision making between clinicians, patients, and their carers. The efficacy of presenting the data in this way fits the scope of the current study but could be the subject of future research.

### 4.1. Limitations

The cognitive tests used in the OCS-Bridge battery have different sensitivities and granularities. The classification of some of the test performances to show intact cognitive function within a given domain may reflect insensitivity of that particular test, especially where count scores determined deficit rather than continuous data such as reaction times. Our study has a considerable limitation in its external validity due to the strict eligibility criteria for participants to undertake the cognitive testing. They needed to be functionally well with a performance status of WHO 0 or 1, be physiologically fit enough for consideration of resection surgery rather biopsy, and have relatively superficial contrast-enhancing tumours for the same. These are therefore the patients in the best functional and physical state for maximal treatment regimens. Those who had a timepoint 3 assessment were required to be local to the Cambridge region for their adjuvant treatment and to have avoided surgical complications which would preclude them from maximal adjuvant therapy. These patient selection and geographical biases limit the external validity to all patients with glioblastoma, and hence for patients with even more severe and extensive disease, the risks of cognitive deficits are likely to be even higher than described here.

We experienced good retention for follow-up testing owing to patient and public involvement in study design. Only one participant refused follow-up testing due to it being too onerous on one A3 timepoint, however we had participants who were not eligible for assessment at A3 due to suffering surgical complications such as large territory stroke or postoperative seizures. To avoid confounding the dataset with the effects of these additional pathological processes, these patients were excluded post-complication, once again causing the dataset to reflect the performance who fared best through the treatment journey. As is common with studies of less common cancers, the number of cases relative to the number of relevant variables required to address the questions is low, however the effect sizes of the cognitive deficit risks seen still convey statistically significant associations of surgical resection upon the risk of accruing further cognitive deficits for patients with glioblastoma.

This study does not clarify the mechanism of this increase in risk to cognition. An apparent assumption may be of direct surgical injury to functional tissue adjacent to tumour and the subsequent disruption to cognitive network function. The deterioration and recovery our data have shown between A1, A2, and A3 implies a plasticity in cognitive function. It is likely that early postoperative inflammation and hormonal stress responses [45] in the body as participants are in the early recovery at A2 may be a factor in the deficit exacerbation. However, there are mounting data to show that surgery [21,46] and glioblastoma pathophysiology [47,48] can affect function in the contralateral hemisphere to that of the contrast enhancing tumour, raising the possibility of a global stun effect upon the whole brain from surgical trauma to one region of it. Future studies are needed to address this mechanistic question to understand how we can minimise the impact of surgery on deteriorating cognitive function.

### 4.2. Interpretation

Our results completely support the previous studies [49,50,51] in the field and the findings from patient reported outcomes described in charity reports [14], where poor cognitive function is linked to poor quality of life in these groups. Overall, for many tests, irrespective of the group odds changing between A1 and A3, the overall performances remained in the deficit range. This was especially true for the tests where there was higher data granularity and where the cognitive function required diffuse recruitment of multiple brain networks spanning both cerebral hemispheres. These were also the tests with background literature implicating ecological validity in day-to-day activity limitation related to the deficits in other neurological disease contexts. Examples include the social withdrawal associated with emotion recognition deficits [52] and the negative associations between employment and impaired processing speed [53].

For the first time, we showed the ‘natural history’ of the deficits in glioblastoma changing from before surgery to early after surgery and then to before adjuvant chemotherapy and radiotherapy at 4–6 weeks after surgery. Three recent studies focusing on glioblastoma had their earliest post-surgical assessment time at 3 months [18,19,54], so the current study explains the picture in-between. They found that the majority of those that were retained for reassessment at 3 months were ‘stable’ when compared to their performance before surgery. This implies that for those participants, whatever outcome they had at A1, deficit or intact, was similar to that at 3 months postoperatively. Our data have added that in-between these time points there is a postoperative worsening early after surgery at A2, which can then show some improvement and recovery by our A3 timepoint before radiotherapy. In conjunction with the findings of those studies, this stabilising of cognitive performance at 4–6 weeks after surgery (A3) may persist to their postoperative assessment time of 3 months. Computerised batteries were used in both studies and showed high levels of deficit at the group level, whichever timepoint was studied. In a broader context, our findings are also comparable with a landmark study, which found all their high-grade glioma participants to have cognitive deficits (defined as impairment in at least 3 tests in their battery) at the point of eligibility for adjuvant radiotherapy [12]. This would be the equivalent of the A3 timepoint in our study. These results collectively show a baseline risk of cognitive deficits from before surgery, which worsens at A2 (2–8 days postoperatively from our results) and then a reduction in risk towards the preoperative baseline at 4–6 weeks after surgery, just before radiotherapy, followed by a persistence of this cognitive burden risk at 3 months postoperatively.

Another study using meta-analysis [55] also comprised data in the early period after surgery found improvements in cognition, however a major confounding factor is that the study sample comprised a majority of low-grade glioma patients. Therefore, our data are more relevant in the context of glioblastoma, with our data being less skewed by the results of heterogeneous, less aggressive pathologies.

Our findings have implications for the possibility of targeting cognitive rehabilitation on a personalized basis. This could be adjusted on the deficit profile of each participant during this earlier period before adjuvant therapies begin at 6 weeks after surgery to maximize the chance of recovery. This may be especially helpful to coincide with a time when there is a reduction in risk of deficits being suggested, even without such specified rehabilitation, as a normal part of their recovery. In the patient journey, this period is when patients are first discharged home after surgery, aware of the diagnosis and prognosis, are unable to return to work, unable to drive, and hence may be inclined for rehabilitation in the form of on-line tools and teleconferencing, as has been piloted in a paediatric brain cancer study [56] and is normalized in the post-COVID-19 era [57]. The data from this current study have formed the basis of ongoing feasibility studies in this area.

The overall patterns of deficits seen are largely consistent when compared across the patients as grouped by hemisphere or lobe containing the contrast-enhancing component of the tumour. This is supported by other studies whose findings support the widespread dissemination of glioblastoma malignant cells throughout the brain [9,47,48,58,59], as well as the current connectomic [60,61,62] understanding of cognitive brain functions, suggesting that cognitive functions are underpinned by networks spanning the whole brain rather than having isolated lobar localisation. Any notable differences are more likely due to the lack of redundancy from the tumour and surgery affecting hub areas within the network, rather than affecting the locus subserving any given function. Indeed, the finding of praxis deficit risk being greater when the contrast enhancing tumour was located in the left language dominant hemisphere is supported by studies finding similar hemispheric asymmetry in function involving skilled limb movement and intact left–right awareness, with disconnection of the deep white matter in the left parietal lobe postulated as the mechanism [63,64,65].

### 4.3. Generalisability

Whilst the risk of cognitive deficits for all patients with glioblastoma is likely to be higher than studied here, our results are particularly relevant for those patients who, similar to our cohort, have a good functional baseline (WHO 0 or 1), have relatively surgically accessible contrast enhancing lesions for the consideration of resection surgery, and remain eligible for maximal adjuvant treatment with chemotherapy and radiotherapy. The age interquartile range of 54–66 shows the population that this most readily generalises to. Some differences attributable to geographic biases may exist, as may limitations in generalisability to non-English speakers.

Aside from the study results, the methods employed of using computerized cognition testing batteries are very generalizable to other neuro-oncology and general oncology contexts. Such tools have automated comparison to healthy normative control data for rapid analysis of individual performance into normal and deficit categories, as has been demonstrated in other studies [8,18,19,54]. Our patient and public involvement advice prior to the start of this study advised using these tools on presurgical assessment visits for A1 datasets, prior to discharge home after surgery, or before clinic visit for histology results for A2 datasets and alongside radiotherapy planning visits for A3 datasets. Using computerized tools and a similar data collection scheme could generalize to other units and facilitate a high level of recruitment and retention, as we have found.

## 5. Conclusions

Tablet computer cognition assessments revealed that the high risk of cognitive deficits in patients with glioblastoma before surgery is exacerbated in the early period after surgery (<2 weeks) when they are discharged home or seen in the outpatient clinic to discuss histology results. For those patients who have a good performance status and are able to undergo adjuvant chemotherapy and radiotherapy postoperatively, we found that the risks of cognitive deficit can improve towards those seen at the presurgical baseline (~4–6 weeks postoperatively). The burden of these risks was personalized to each participant, and these individual differences in cognitive performances were longitudinally detectable with our methods.

Future research is required to further validate our findings. Furthermore, based on the current findings, future research could explore mechanistic understanding at the network or cellular level, which underpins the recovery suggested by deficit risk reduction between 2 to 6 weeks after surgery. Finally, other mechanism agnostic research may be targeted in this timeframe to import efficacious tools proven in other neurological contexts to provide targeted and personalized cognitive rehabilitation in this population.

## Figures and Tables

**Figure 1 jpm-13-00278-f001:**
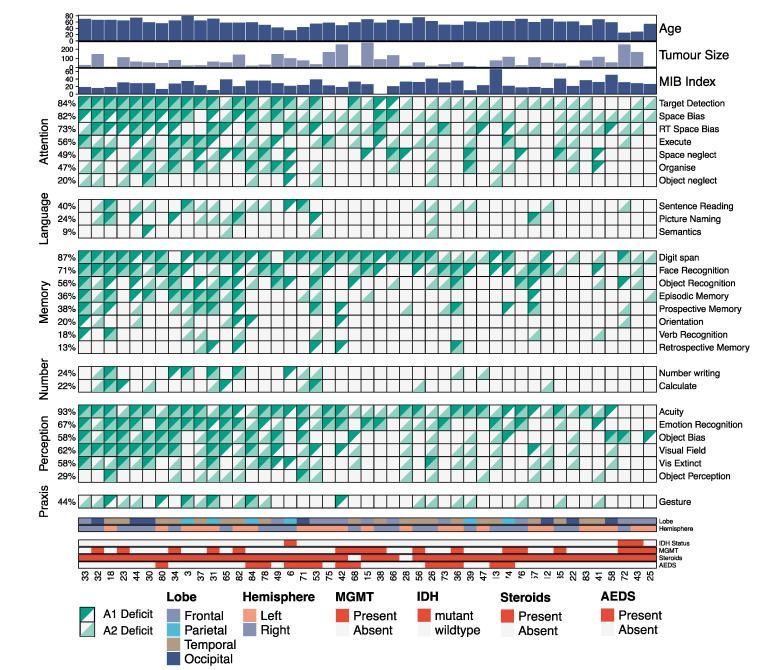
Summary of cohort characteristics. Each column represents one patient. For the heatmap, each square represents the cognitive status of a patient at A1 (top left triangle) and A2 (bottom right triangle). Information relative to the clinical characteristics of each patient is annotated at the top and bottom of the heatmap.

**Figure 2 jpm-13-00278-f002:**
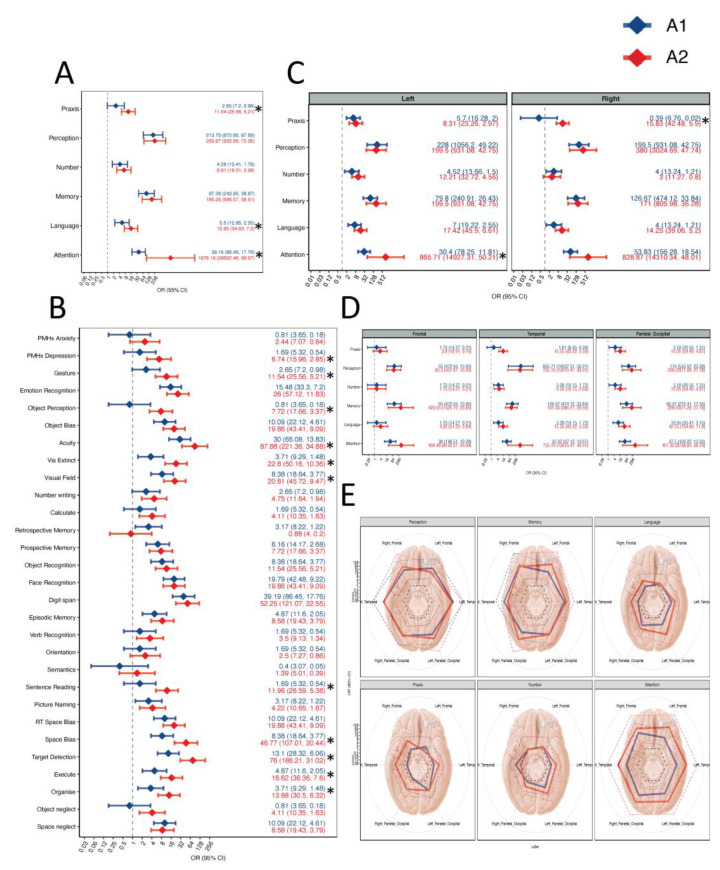
Risk of cognitive deficit at A1 and A2. (**A**) Forest plot displaying the OR of cognitive deficit at A1 and A2, relative to A0 (normative risk), for each cognitive domain across the entire cohort. (**B**) Forest plot displaying the OR of cognitive deficit at A1 and A2, healthy control risk (dotted line) for each cognitive test across the entire cohort. (**C**) Forest plot displaying the OR of cognitive deficit at A1 and A2, healthy control risk (dotted line) for each cognitive domain, stratified by hemispheric location of the lesion. (**D**) Forest plot displaying the OR of cognitive deficit at A1 and A2, healthy control risk (dotted line), for each cognitive domain, stratified by lobar location of the lesion. (**E**) Radar plot displaying the OR of cognitive deficit at A1 and A2, healthy control risk (dotted line), for each cognitive domain, stratified by location (hemisphere and lobe) of the lesion.

**Figure 3 jpm-13-00278-f003:**
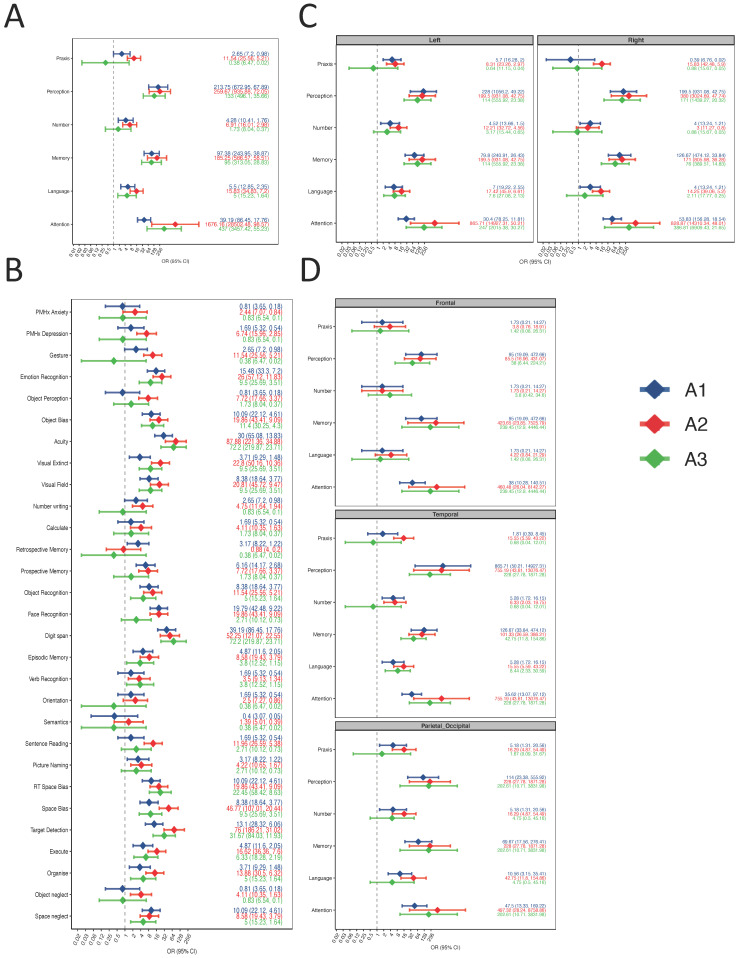
Risk of cognitive deficit at A1, A2, and A3. (**A**) Forest plot displaying the OR of cognitive deficit at A1, A2, and A3, relative to healthy control risk (dotted line), for each cognitive domain across the entire cohort. (**B**) Forest plot displaying the OR of cognitive deficit at A1, A2, and A3, relative to healthy control risk (dotted line), for each cognitive test across the entire cohort. (**C**) Forest plot displaying the OR of cognitive deficit at A1, A2, and A3, relative to healthy control risk (dotted line), for each cognitive domain, stratified by hemispheric location of the lesion. (**D**) Forest plot displaying the OR of cognitive deficit at A1, A2, and A3, relative to healthy control risk (dotted line), for each cognitive domain, stratified by lobar location of the lesion.

**Table 1 jpm-13-00278-t001:** Multivariate regression model of the association between each characteristic and cognitive deficit at A1.

		Tumour Volume (cm^3^)	Age at Surgery (Years)	IDH Status (Mutant)	MIB Index (%)	MGMT Methylation Status (Y/N)	Steroids Pre-operatively (Y/N)	AEDS1 (Y/N)	COVID Era (Y/N)	Age Left Education (Years)
**Overall**		75.6 (22.4–108.9)	58 (54–66)	6%	28.0 (19.5–33.0)	40%	90%	31%	27%	18 (16–21)
	Deficit	74.3 (5.6–268.1)	58.3 (25–78)	0 (0–1)	28.6 (5–65)	0.4 (0–1)	0.9 (0–1)	0.3 (0–1)	0.1 (0–1)	17.8 (14–23)
**Perception**	No Deficit	89.4 (14.1–164.6)	56 (28–69)	0.2 (0–1)	20.9 (15–28)	0.2 (0–1)	0.8 (0–1)	0 (0–0)	0 (0–0)	19.8 (16–26)
	*p.val*	*0.70*	*0.83*	*0.23*	*0.05*	*0.64*	*0.36*	*0.29*	*1.00*	*0.49*
	Deficit	80.2 (6.4–268.1)	59.2 (25–78)	0 (0–1)	27.5 (5–65)	0.4 (0–1)	0.9 (0–1)	0.3 (0–1)	0 (0–1)	17.8 (14–23)
**Memory**	No Deficit	51.6 (5.6–164.6)	52.1 (28–61)	0.1 (0–1)	30.2 (18–50)	0.5 (0–1)	0.9 (0–1)	0.1 (0–1)	0.2 (0–1)	19.2 (16–26)
	*p.val*	*0.20*	*0.12*	*0.42*	*0.55*	*0.70*	*1.00*	*0.40*	*0.12*	*0.30*
	Deficit	67.3 (6.4–137.3)	58.4 (33–78)	0.1 (0–1)	23.7 (5–38)	0.3 (0–1)	0.9 (0–1)	0.3 (0–1)	0.1 (0–1)	18.9 (15–23)
**Language**	No Deficit	78 (5.6–268.1)	58 (25–74)	0.1 (0–1)	29.2 (9.2–65)	0.4 (0–1)	0.9 (0–1)	0.3 (0–1)	0.1 (0–1)	17.7 (14–26)
	*p.val*	*0.52*	*0.93*	*0.54*	*0.12*	*0.48*	*1.00*	*1.00*	*1.00*	*0.30*
	Deficit	66.2 (6.4–248.5)	62.7 (49–78)	0 (0–0)	21.2 (10–35)	0.3 (0–1)	1 (1–1)	0.5 (0–1)	0.2 (0–1)	17 (15–19)
**Praxis**	No Deficit	76.9 (5.6–268.1)	57.4 (25–74)	0.1 (0–1)	28.9 (5–65)	0.4 (0–1)	0.9 (0–1)	0.3 (0–1)	0.1 (0–1)	18.1 (14–26)
	*p.val*	*0.79*	*0.30*	*1.00*	*0.15*	*1.00*	*1.00*	*0.37*	*0.42*	*0.15*
	Deficit	65.4 (6.4–137.3)	60.3 (33–78)	0.1 (0–1)	26.2 (10–38)	0.4 (0–1)	1 (1–1)	0.2 (0–1)	0.1 (0–1)	17.2 (15–23)
**Number**	No Deficit	77.8 (5.6–268.1)	57.6 (25–74)	0 (0–1)	28.4 (5–65)	0.4 (0–1)	0.9 (0–1)	0.3 (0–1)	0.1 (0–1)	18.2 (14–26)
	*p.val*	*0.50*	*0.55*	*0.46*	*0.59*	*1.00*	*0.57*	*1.00*	*0.57*	*0.35*
	Deficit	70.5 (5.6-268.1)	60.2 (33-78)	0 (0–1)	27.3 (9.2–62)	0.3 (0–1)	0.9 (0–1)	0.3 (0–1)	0.1 (0–1)	17.8 (14–23)
**Attention**	No Deficit	86.1 (11.5-248.5)	53.7 (25–74)	0.1 (0–1)	29.3 (5–65)	0.6 (0–1)	0.9 (0–1)	0.4 (0–1)	0.1 (0–1)	18.5 (15–26)
	*p.val*	*0.47*	*0.09*	*0.25*	*0.64*	*0.19*	*1.00*	*0.32*	*1.00*	*0.46*

All *p*-values are shown as unadjusted. *p*-value < 0.05 is considered statistically significant.

**Table 2 jpm-13-00278-t002:** Multivariate regression model of the association between each characteristic and cognitive deficit at A1, A2, and ΔA.

Covariate	Perception			Memory			Language			Praxis			Number			Attention		
	A1	A2	ΔA	A1	A2	ΔA	A1	A2	ΔA	A1	A2	ΔA	A1	A2	ΔA	A1	A2	ΔA
Tumour Volume (cm^3^)	0.99	1.00	1.00	0.12	1.00	1.00	0.52	0.63	0.15	0.47	1.00	0.11	0.13	0.27	0.46	0.97	1.00	0.97
Age at A1 (years)	0.99	1.00	1.00	0.16	1.00	1.00	0.29	0.37	0.17	0.07	1.00	0.77	0.30	0.32	0.30	0.19	1.00	0.11
IDH status (mutant)	1.00	1.00	1.00	0.56	1.00	1.00	0.18	0.22	0.37	1.00	1.00	1.00	0.06	1.00	1.00	0.78	1.00	0.45
MIB index (%)	0.99	1.00	1.00	0.05	1.00	1.00	0.42	0.77	0.28	0.04	1.00	0.47	0.65	0.92	0.64	0.79	1.00	0.43
MGMT Methylation Status (Y/N)	0.99	1.00	1.00	0.10	1.00	1.00	0.17	0.56	0.41	0.27	1.00	0.13	0.61	0.89	0.20	0.07	1.00	0.14
Steroids pre-operatively	1.00	1.00	1.00	0.19	1.00	1.00	0.86	0.99	0.99	1.00	1.00	1.00	0.99	1.00	1.00	0.34	1.00	1.00
AEDS1	1.00	1.00	1.00	0.17	1.00	1.00	0.58	0.26	0.44	0.06	1.00	0.73	0.82	0.17	0.80	0.94	1.00	0.23
COVID era	0.99	1.00	1.00	0.92	1.00	1.00	0.99	0.50	0.91	0.20	1.00	0.91	1.00	1.00	1.00	0.99	1.00	0.71
Age Left Education (years)	0.99	1.00	1.00	0.52	1.00	1.00	0.28	0.51	0.09	0.67	1.00	0.25	0.49	0.41	0.92	0.40	1.00	0.87

Multivariate regression model of the association between each characteristic and cognitive deficit at A1, A2, and ΔA. For conciseness, only *p*-values and no effect sizes are shown here. The entirety of the model and all its parameters are available in Appendix A. Appendix A contain the univariate study of association between each covariate and cognitive deficit. All *p*-values are shown as unadjusted. *p*-value < 0.05 is considered statistically significant. A1: Assessment of cognitive deficit before surgery, A2: Assessment of cognitive deficit in the early postoperative period, ΔA: Onset of cognitive deficit between A1 and A2.

## Data Availability

For data sharing and prevention of research waste, the raw data used to determine the odds risks presented are included as Appendix A).

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
