# Peer review of "A Prospective Study of Longitudinal Risks of Cognitive Deficit for People Undergoing Glioblastoma Surgery Using a Tablet Computer Cognition Testing Battery: Towards Personalized Understanding of Risks to Cognitive Function"

_jpm, 2023, doi:10.3390/jpm13020278_

Round 1

Reviewer 1 Report

This analysis is of great interest and really important for better patient compliance if physicians could therefore better address their cognitive interests and inquiries. 

Here I have a couple of follow-up questions in terms of content: 

1) Please elaborate a bit on this sentence in the Introduction (39-40): "(...) the subject of patient reported outcome literature." 

2) Could you explain why you used " OCS-54 Bridge" as a cognitive test battery? 

3) Please elaborate in your Discussion: In general, what about patients who underwent brain surgery and eventually recovered? Here, a comparison in the literature would be interesting, especially regarding the time post-op in patients after brain surgery and their cognitive development. Could you elaborate further here?

Reviewer 2 Report

This is a unique study of evaluation of cognitive functions before and after glioma surgery and postoperative adjuvant radiochemotherapy.

Cognitive functions are occasionally altered after not only surgery but also radiochemotherapy.

Given Broadman’s map representing the functional location in the cerebral hemisphere, I agree results of the current manuscript describing that praxis deficit was significant greater in patients with GBM affecting left hemisphere, whereas memory, number, and perception had similar risk of deficits irrespective of hemisphere.

I would accept results of correlation analyses that age was associated with praxis and memory. Interestingly, IDH-wild type status was associated with perception deficit. I do not expect that genetic status of the tumor was associated with cognitive function of patients.

As authors mentioned in the discussion section, I thought that steroid and anti-epileptic agents might be confounding factors.

Those results were unexpected.

I would expect that further studies regarding lower grade gliomas would be interesting.

Author Response

Comment 1: 'This is a unique study of evaluation of cognitive functions before and after glioma surgery and postoperative adjuvant radiochemotherapy.'

We are very grateful for this comment about our study. 

Comment 2: 'Cognitive functions are occasionally altered after not only surgery but also radiochemotherapy.'

We agree with this and mention it in lines 46-50.

Comment 3: 'Given Broadman’s map representing the functional location in the cerebral hemisphere, I agree results of the current manuscript describing that praxis deficit was significant greater in patients with GBM affecting left hemisphere, whereas memory, number, and perception had similar risk of deficits irrespective of hemisphere.'

We are pleased that our interpretation of the data and literature findings is in accordance with the reviewer.

Comment 4: 'I would accept results of correlation analyses that age was associated with praxis and memory. Interestingly, IDH-wild type status was associated with perception deficit. I do not expect that genetic status of the tumor was associated with cognitive function of patients.'

These weak correlative associations were lost in the multi-variate analyses (lines 325-329), so they are likely to be artefact or confounded from hidden explanatory variables. We agree with the reviewer and also do not expect this to be a biologically based association.

Comment 5: 'As authors mentioned in the discussion section, I thought that steroid and anti-epileptic agents might be confounding factors. Those results were unexpected.'

We had also hypothesised them to be confounding factors, hence their inclusion in the study. However their effect size was too small to be detected with the sample size in this study when compared with larger effects such as undergoing neurosurgery (lines 381-385).

Comment 6: 'I would expect that further studies regarding lower grade gliomas would be interesting.'

We thoroughly agree and are working on them currently.